# Spatial Learning by Using Non-Visual Geometry and a Visual 3D Landmark in Zebrafish (*Danio rerio*)

**DOI:** 10.3390/ani13030440

**Published:** 2023-01-27

**Authors:** Greta Baratti, Sara Boffelli, Davide Potrich, Valeria Anna Sovrano

**Affiliations:** 1CIMeC, Center for Mind/Brain Sciences, University of Trento, 38068 Rovereto, Italy; 2Department of Psychology and Cognitive Science, University of Trento, 38068 Rovereto, Italy

**Keywords:** spatial cognition, reorientation, geometric module, environmental geometry, transparency, 3D landmark, sensory channels, multisensory, zebrafish

## Abstract

**Simple Summary:**

Orienting in space requires building informative relationships based on environmental cues, which animals use to meet survival needs. During navigation, spatial geometries interact with featural information, thereby providing a cohesive representation where a variety of concurrent elements must be considered. When a conspicuous landmark marks a distinctive geometric space, the ability to use or even conjoin both is essential, most importantly, in conditions of visual deprivation. The present study investigated whether zebrafish (*Danio rerio*) reoriented, taking advantage of the transparent geometry of a bounded rectangular arena equipped with a blue-colored 3D landmark, for finding food and social rewards. Fish were required to reorient over time while varying the landmark’s position (i.e., proximal/distal; on the short/long arena’s side) in relation to a target corner. The repeated experience would have allowed fish to handle the two pieces of information which depend on different sensory pathways: the visual system, as regards the landmark; extra-visual systems (e.g., haptic and tactile-like), as regards geometry. Results revealed zebrafish’s reorientation behavior ability within the non-visual geometric framework and the use of the visual 3D landmarks, which are partially affected by proximity–length interactions. For the first time in zebrafish, we report the possible recruitment of different sensory channels (sight and touch, i.e., “wall-following” exploration of boundaries) for experiencing multi-factor life environments.

**Abstract:**

Fish conjoin environmental geometry with conspicuous landmarks to reorient towards foraging sites and social stimuli. Zebrafish (*Danio rerio*) can merge a rectangular opaque arena with a 2D landmark (a blue-colored wall) but cannot merge a rectangular transparent arena with a 3D landmark (a blue cylinder) without training to “feel” the environment thanks to other-than-sight pathways. Thus, their success is linked to tasks differences (spontaneous vs. rewarded). This study explored the reorientation behavior of zebrafish within a rectangular transparent arena, with a blue cylinder outside, proximal to/distal from a target corner position, on the short/long side of the arena. Adult males were extensively trained to distinguish the correct corner from the rotational one, sharing an equivalent metric–sense relationship (short surface left, long surface right), to access food and companions. Results showed that zebrafish’s reorientation behavior was driven by both the non-visual geometry and the visual landmark, partially depending on the landmark’s proximity and surface length. Better accuracy was attained when the landmark was proximal to the target corner. When long-term experience was allowed, zebrafish handled non-visual and visual sensory stimulations over time for reorienting. We advance the possibility that multisensory processes affect fish’s reorientation behavior and spatial learning, providing a link through which to investigate animals’ exploratory strategies to face situations of visual deprivation or impairments.

## 1. Introduction

Animals use environmental geometry as a reference scheme to localize current positions and search for salient objects (reorientation behavior). In natural contexts, geometric cues refer to invariant parameters of terrain-like surfaces, which therefore acquire ecological meanings [1]. In controlled contexts, geometry use for reorienting has been experimentally tested by using rooms or arenas characterized by informative geometries, such as kite-shaped, parallelogram and rectangular ones, with the aim of reproducing spatial relationships between large-scale metrics and environmental landmarks (reviewed in [2]).

Integrating rectangular geometry with conspicuous landmarks has been a debated issue in spatial reorientation research. From Cheng’s studies [3,4] and more recent investigations (reviewed in [5,6,7,8,9]), in vertebrate species, the process seems to interact with several intervening factors, such as the type of task (spontaneous choices vs. rewarded training), the environment’s size (large vs. small), the opportunity to move within the environment (manipulation vs. locomotion), the landmarks’ proximity (proximal vs. distal) and salience. Moreover, the evidence does not even converge on the mechanisms behind reorientation behavior. There is a variety of views, such as view-based mechanisms [10,11,12,13,14], modular codes [9], exploratory strategies based on movement [15,16,17,18] and other-than-sight pathways [19,20,21,22].

The zebrafish (*Danio rerio*) is a limnophilic cyprinid widely used as a model in biological sciences [23,24,25,26] whose behavior in spatial cognition domains has been investigated for a long time [27,28,29]. In geometric reorientation, the zebrafish has also exhibited good performance and precision while reorienting in different experimental situations (reviewed in [5]). Previous studies have shown that zebrafish reorient within a rectangular opaque arena, if observed in spontaneous unrewarded tasks [30,31,32] or unspontaneous rewarded tasks [33]. Moreover, zebrafish learned to reorient if observed, under extensive training, within a rectangular transparent arena [22], and thus, in conditions of spatial visual deprivation. It has been reported that zebrafish spontaneously integrated the shape of a rectangular opaque arena with a conspicuous 2D landmark (a blue-colored surface, also called the “blue wall” task [30], which is widely documented for other vertebrate groups, and more recently in humans [34,35,36,37,38]) but did not integrate the shape of a rectangular transparent arena with a conspicuous 3D landmark (that is, a blue-colored cylinder), in the absence of training [32]. In detail, the 3D landmark was placed outside, midway one side of the transparent arena, and its proximity to a target corner position (proximal vs. distal) could vary in relation to the lengths of surfaces (short vs. long side) to give four combinations (Proximal–Short (PS); Proximal–Long (PL); Distal–Short (DS); Distal–Long (DL)). This work reported that zebrafish failed at reorienting when the 3D landmark was: (1) near the target corner position on the long side of the arena—PL condition; (2) far from the target corner position on the short side of the arena—DS condition; (3) far from the target corner on the long side of the arena—DL condition. Only when the landmark was near the target corner position on the short side of the arena—PS condition—did zebrafish exhibit a preference for the target corner position, while being biased by the closest corner position, though. In other words, fish distinguished the correct from the rotational location (i.e., resolved the spatial symmetry), but also preferred the corner location nearest the correct one since it was equally marked by the landmark. Such behavior has been explained in terms of landmark-perceptive salience and attractiveness, where a primacy visual cue prevents the use of spatial geometry for reorienting. However, another view suggests that the type of behavioral task affects reorientation abilities, as it has been documented in other studies (mice: [39]; zebrafish: [40]; redtail splitfin fish: [22].

Spontaneous unrewarded and unspontaneous rewarded tasks have been validated well to test the reorientation behavior of teleost fish (reviewed in [5]) and differ in several aspects. The first task, the “social-cued memory task”, consists of spontaneously reorienting over a limited number of trials in one single test session. Fish are required to find the position of a no-longer-present social reward after “passive disorientation”, which consists in turning on place the fish (usually confined within a jar) through a 360° clockwise/counter-clockwise rotation to wipe ego-centered reference coordinates. The behavior is not rewarded; thus, it seems mostly driven by attentional factors [22,30,31,32]. The second task, the “rewarded exit task”, consists in reorienting over time on a trial-and-error basis. Fish are required to choose the symmetrical corner positions, leading to exiting the experimental arena and reaching food and social companions. The behavior is rewarded; thus, it seems mostly driven by motivational states [21,22,33,40,41,42,43,44,45,46].

With the aim of understanding whether zebrafish failing at spontaneously using or even conjoining non-visual geometry and visual landmarks was due to cognitive limits or the tasks’ requests, in the present study, we reproduced the experimental conditions just described [32], but extensively trained fish through the rewarded exit task. If a kind of repeated experience enhances spatial geometric representations, as already suggested [22,40,46], we would expect more efficient reorientation at the expense of attractiveness biases. It was even possible that landmark proximity and surface length could somehow affect the integration process: in such a case, an increasing range of difficulty in learning (where: Proximal–Short > Proximal–Long > Distal–Short > Distal–Long) could be observed.

## 2. Materials and Methods

Subjects were 16 adult male zebrafish (wild type), ranging from 3 to 5 cm in body-length, which came from breeding stocks of our laboratory (CIMeC, University of Trento). Fish were hosted within familiar home tanks (Wave Zen Artist, Amtra^®^, 35 × 28 × 30 cm, 27 L capacity), in which a hang-on-back filter (Niagara 250, WAVE) ensured the quality of water and a 50 W heater (Newa Therm^®^, NEWA^®,^ Loreggia, Italy) maintained the temperature of the water at 26 ± 1 °C. Fish were reared in a light–dark 14:10 photoperiod, and before starting the experiment, they were fed twice a day by administering dry food (Vipan, sera^®^, Eindhoven, The Netherlands).

The apparatus was the same as used in previous studies [40,46], consisting of a rectangular glass arena (30 × 20 × 8 cm) made of two long (26 × 11 cm) and two short (16 × 11 cm) walls, which were placed on a PVC basement (50 × 50 cm) and covered with dark monochromatic gravel (3 cm in depth). At the level of each corner, at 2.5 cm facing outward, there was a transparent vertical “corridor” composed of three parts: one glass sheet (central part: 3 × 11 cm) and two acetate sheets (lateral parts: 2.5 × 9 cm); these acetate sheets were perpendicularly glued on the glass sheet and were characterized by a different pattern of vertical slits. The rewarding corridors had a thick central slit (1 × 7.43 cm) and two thin lateral slits (0.2 × 7.43 cm), allowing fish to swim through and leave the arena. Conversely, the unrewarding corridors had a 3 × 3 matrix of thin slits (upper and lower series: 0.3 × 2.5 cm; central series: 0.3 × 2 cm), forcing fish to remain inside the arena. The slits’ overall perimeter was balanced (47.4 cm) to exclude hydrodynamic effects detectable by tactile-like systems, such as the lateral line [21,22,47,48,49]. The corridors were allocated at the corners, upright, and diagonally aligned with the arena’s walls. The glass arena was allocated within a circular plastic amaranth tank (diameter × height: 175 × 27 cm), which was surrounded by a circular black curtain fixed on a wood-and-metal frame. It was homogeneously lit from above (height: 100 cm) through a 24-watt fluorescent white light tube (Lumilux, Osram GmbH, D, Munich, Germany) since the apparatus was hosted in a darkened room. To create a perceptive continuity without light reflections, the arena was submerged (0.5 cm gap in height). The water temperature was kept around 26 ± 1 °C by a 50-watt heater (NEWA Therm^®^, NEWA, Loreggia, Italy), and a filter (NEWA Duetto^®^, NEWA, Loreggia, Italy) made sure of good water quality (both the filter and heater were removed during the experiment). In the center of the arena, there was a glass cylinder (diameter × height: 6 × 8 cm) hosting the experimental fish, which could be vertically lifted through a pulley mechanism installed on the frame’s upper end, releasing the fish. Fish behavior was video recorded from above by means of a webcam (LifeCam Studio, Microsoft, I, Washington, DC, USA). See Figure 1.

The landmark was a plastic cylinder (diameter: 9.5 cm; height: 14 cm) externally covered with a blue plastic sheet (RGB: 30, 144, 255).

Four experimental conditions were designed (N = 4 each), which differed by landmark position in relation to the correct corner position and the arenas’ side: in the Proximal–Short (PS) condition, the landmark was placed near the correct corner position (from now labelled as “Correct” or “C”) on the short side of the arena; in the Proximal–Long (PL) condition, the landmark was placed near the Correct on the long side of the arena; in the Distal–Short (DS) condition, the landmark was placed far from the Correct on the short side of the arena; in Distal–Long condition (DL), the landmark was placed far from the Correct on the long side of the arena. See Figure 2.

The procedure of the rewarded exit task was the same as that validated in previous studies [21,22,33,40,41,42,43,44,45,46] and basically consisted in a training over time to meet an accuracy threshold ≥ 70% in two sessions (labelled as: “learning”: the fish meets the threshold; “validation”: the fish confirms the threshold), which defined that learning had been achieved. In detail, the criterion required that an accuracy ≥ 70% for the correct corner position must be obtained in two consecutive training sessions (i.e., in two consecutive days of observation). Instead of the 10 training sessions provided in the previous studies, here we decided to give all the fish 25 training sessions at maximum for learning, one per day, since possible effects due to Proximity–Length interactions might be expected. Each session had 8 consecutive trials, whose overall duration could vary between 60 and 90 min, depending on the fish’s performance. Since each fish was individually observed within the experimental apparatus, 4–5 individuals at most in a day took part in the training. At the beginning of the session, the experimental fish was gently moved from its home tank to the apparatus, within the annular region outside the testing arena, and it was allowed to get familiar with the environment for 5 min. After that time, the fish was moved within the glass cylinder placed in the center of the arena for 30 s. Then, the cylinder was lifted and the fish was free to explore the arena and approach the vertical corridors at the four corners. The fish could behave in different ways, choosing: (1) the correct corner, as its first attempt (full reinforcement: food, two female companions, 6 min rest-time); (2) C, after having approached the incorrect corners, i.e., R, N, or F (no reinforcement: 2 min rest-time); (3) no corners within the 10 min provided for the trial (null trial, no reinforcement: 5 min rest-time). In this last case, three subsequent null trials led to stopping the training session until the next day. Valid choices required the fish to enter the corridor at least with 2/3 of its body-length and show tail movements to force an escape. Before starting each new trial, the tester turned the apparatus 90° clockwise to prevent the fish from using ego-centered coordinates or local cues to locate the correct corner position.

The following dependent variables were measured: the mean number of trials (𝑓) in the experimental conditions where fish met the learning criterion of ≥ 70% accuracy towards the Correct; total choices (𝑓) towards the four corners of the experimental arena; total choices (𝑓) towards the Correct with respect to motion patterns (motion strategy: wall-following vs. center-to-corner; motion direction: left vs. right); latency times (s) before going out of the arena.

Motion patterns were calculated as proportion indexes through the formulas:$$Motion\;Strategy\;Index=\frac{C_{W}}{\left(C_{W}+{\;C}_{C}\right)}$$
$$Motion\;Direction\;Index=\frac{C_{\mathrm{WL}}}{\left(C_{\mathrm{WL}}+{\;C}_{\mathrm{WR}}\right)}$$

“C_W_” = correct wall-following; “C_C_” = correct center-to-corner; “C_WL_” = correct wall-following left; “C_WR_” = correct wall-following right.

A repeated-measures ANOVA was applied to compare the total choices towards the four corners (correct “C”, near “N”, rotational “R”, far “F”) and the two diagonals (correct “CR”, incorrect “NF”), depending on the experimental condition (Proximal–Short “PS”, Proximal–Long “PL”, Distal–Short “DS”, Distal–Long “DL”) and session. A repeated-measures ANOVA was also applied to motion strategy, motion direction, and latency times. Bonferroni’s test was performed for analyzing post hoc multiple comparisons. The Shapiro–Wilk test was performed to assess normality, whereas Levene’s test of equality of error variances and the Mauchly’s sphericity test were performed to assess homoscedasticity. To estimate the effect size of significant data analysis, we reported $\eta_{p}^{2}$
as an index for ANOVA. Statistical analyses were performed with IBM^®^ SPSS Statistic 27 software package and row data are reported in a Appendix A .

## 3. Results

A post hoc power analysis was performed, showing that the sample of 16 animals (4 per each experimental condition) allowed us to detect a moderate effect size of 0.77, with a power of 0.72. A similar sample was used in a previous study [46] about the role of environmental geometry and learning processes on landmark-use in the teleost fish *Xenotoca eiseni*.

### 3.1. Individual Learning Curves

The individual learning curves for each experimental condition (PS, PL, DS, DL) are shown in Figure 3, considering the total choices (%) made by fish towards the Correct, as a measure of accuracy. Performance ≥ 70% was obtained in the Proximal–Short (4/4 fish), Proximal–Long (3/4), and Distal–Short (1/4) conditions. No fish met the learning criterion in the Distal–Long condition.

### 3.2. Corners by Conditions (PS, PL, DS, DL) in the Last Four Training Sessions

The total choices (𝑓) made by fish towards the four corners (C, N, R, F) were analyzed to evaluate reorientation behavior within the non-visual geometric frame (rectangular transparent arena) equipped with the visual 3D landmark (blue cylinder) in the training sessions that all the fish had in common (the last four). Note that to compare the choices towards the corners, the same number of sessions that all the fish (successful and not) had in common was needed. Since one of the animals learned in just four sessions, the last four training sessions were analyzed, including the learning and validation sessions, plus the last two before learning for successful fish and from session 22 to session 25 for unsuccessful fish.

A repeated-measures ANOVA was performed by considering the total choices (𝑓). The ANOVA with session (the last four training sessions) and corner (C, N, R, F) as within-subject factors, and condition (PS, PL, DS, DL) as the between-subject factor, showed the following results: There were significant effects of session (*F*_(3,36)_ = 3.07, *p* = 0.04,
$\eta_{p}^{2}$ = 0.2), corner (*F*_(2,24)_ = 69.88, *p* < 0.001, $\eta_{p}^{2}$ = 0.85), corner ∗ condition (*F*_(6,24)_ = 4.5, *p* = 0.003, $\eta_{p}^{2}$ = 0.53), and condition (*F*_(3,12)_ = 28.38, *p* < 0.001, $\eta_{p}^{2}$ = 0.88). There was no significant effect of session ∗ condition (*F*_(9,36)_ = 1.28, *p* = 0.28), session ∗ corner (*F*_(4,51)_ = 1.7, *p* = 0.16), or session ∗ corner ∗ condition (*F*_(13,51)_ = 1.43, *p* = 0.18).

Since there was a difference among conditions, a repeated-measures ANOVA and post hoc multiple comparisons were performed on the total choices (𝑓) for each condition separately.

In the Proximal–Short condition, the ANOVA with session and corner as within-subject factors showed the following results: There were significant effects of corner (*F*_(3,9)_ = 27.28, *p* = 0.01, $\eta_{p}^{2}$ = 0.9) and session (*F*_(3,9)_ = 5.41, *p* = 0.02, $\eta_{p}^{2}$ = 0.64). There was no significant effect of session ∗ corner (*F*_(9,27)_ = 1.64, p = 0.16). Post hoc multiple comparisons with Bonferroni’s correction revealed significant differences between C and N, C and R, and C and F (*p* < 0.001), but not between N and R, N and F, and R and F (*p* = 1).

In the Proximal–Long condition, the ANOVA with session and corner as within-subject factors showed a significant effect of corner (*F*_(3,9)_ = 150.20, *p* < 0.001, $\eta_{p}^{2}$ = 0.98). There was no significant effect of session (*F*_(3,9)_ = 0.78, *p* = 0.53) or session ∗ corner (*F*_(9,27)_ = 1.18, *p* = 0.35). Post hoc multiple comparisons with Bonferroni’s correction revealed significant differences between C and N, C and R, and C and F (*p* < 0.001), but not between N and R, N and F, and R and F (*p* = 1).

In the Distal–Short condition, the ANOVA with session and corner as within-subject factors showed the following results: There were significant effects of corner (*F*_(3,9)_ = 21.38, *p* < 0.001, $\eta_{p}^{2}$ = 0.88). There was no significant effect of session (*F*_(3,9)_ = 1.72, *p* = 0.23) or session ∗ corner (*F*_(9,27)_ = 1.23, *p* = 0.32). Post hoc multiple comparisons with Bonferroni’s correction revealed significant differences between C and N (*p* < 0.001), C and R (*p* < 0.001), and C and F (*p* = 0.003), but not between N and R (*p* = 1), N and F (*p* = 1), and R and F (*p* = 0.35).

In the Distal–Long condition, the ANOVA with session and corner as within-subject factors showed the following results: there was no significant effect of session (*F*_(3,9)_ = 1.11, *p* = 0.4), corner (*F*_(3,9)_ = 0.91, *p* = 0.47), or session ∗ corner (*F*_(9,27)_ = 1.59, *p* = 0.17).

In Proximal–Short, Proximal–Long, and Distal–Short conditions, fish chose the correct corner position (C) more than the incorrect ones (N, R, F) in the last four training sessions. However, depending on the condition, not all the fish met the learning criterion.

### 3.3. Corners in Successful Fish

The total choices (𝑓) made by fish towards the four corners (C, N, R, F) were analyzed after collapsing the three experimental conditions in which fish learned to reorient within the non-visual geometric frame (rectangular transparent arena) equipped with the visual 3D landmark (blue cylinder).

Fish learned to reorient in 119.13 ± 20.65 trials (≈ 17 training sessions). However, the mean number of trials significantly differed between PS and PL (PS: 78.5 ± 25.97; PL: 165.33 ± 19.23; *t*_(5)_ = −2.5, *p* = 0.05, 95% CI [−3.75, 0.03]).

A repeated-measures ANOVA was performed by considering the total choices (𝑓) and including in the analysis eight fish (4 in PS, 3 in PL, 1 in DS). Results are shown in Figure 4. The ANOVA with session (the sessions of learning and validation) and corner (C, N, R, F) as within-subject factors, and condition (PS, PL, DS) as the between-subject factor, showed the following results: There were significant effects of corner (*F*_(1,6)_ = 60.01, *p* < 0.001, $\eta_{p}^{2}$ = 0.92) and session (*F*_(3,15)_ = 7.37, *p* = 0.003, $\eta_{p}^{2}$ = 0.6). There was no significant effect of session ∗ condition (*F*_(6,15)_ = 1.1, *p* = 0.41), corner ∗ condition (*F*_(3,6)_ = 0.66, *p* = 0.58), session ∗ corner (*F*_(9,45)_ =1.87, *p* = 0.08), or condition (*F*_(2,5)_ = 0.97, *p* = 0.44). Post hoc multiple comparisons with Bonferroni’s correction revealed significant differences between C and N, C and R, and C and F (*p* < 0.001), but not between N and R, N and F, and R and F (*p* = 1).

In order to estimate learning performance, the first choices made by fish towards the four corners were considered (even if not analyzed). Percentages in both learning and validation stages are reported in Table 1.

### 3.4. Diagonals in Successful and Unsuccessful Fish

The total choices (𝑓) made by fish towards the two diagonals (correct (CR), incorrect (NF)) were analyzed to evaluate the use of the non-visual geometric frame (rectangular transparent arena) alone, despite the visual 3D landmark (blue cylinder), by successful fish at reorienting. Note that the use of geometry alone was evaluated in those fish that learned to choose the correct diagonal more than or equal to 70% of the time, if considering the sessions of learning and validation (i.e., the last two of training). In such a case, the preference for the correct diagonal, which is calculated by collapsing the correct and the rotational corner positions, would be biased by the preference for the correct one: hence, the two sessions before learning were analyzed. The three experimental conditions in which fish learned were collapsed; mean performance is reported. Results are shown in Figure 5a.

A repeated-measures ANOVA was performed by considering the total choices (𝑓) and including in the analysis eight fish (4 in PS, 3 in PL, 1 in DS). The ANOVA with session (the last two sessions before learning) and diagonal (correct (CR), incorrect (NF)) as within-subject factors, and condition (PS, PL, DS) as the between-subject factor, showed the following results: There was a significant effect of diagonal (*F*_(1,5)_ = 13.25, *p* = 0.02, $\eta_{p}^{2}$ = 0.73). There was no significant effect of session (*F*_(1,5)_ = 4.69, *p* = 0.08), session ∗ condition (*F*_(2,5)_ = 1.20, *p* = 0.38), diagonal ∗ condition (*F*_(2,5)_ = 1.09, *p* = 0.4), session ∗ diagonal (*F*_(1,5)_ = 1.53, *p* = 0.27), session ∗ diagonal ∗ condition (*F*_(2,5)_ = 0.48, *p* = 0.64), or condition (*F*_(2,5)_ = 1.06, *p* = 0.41).

In the experimental conditions where almost all the fish met the learning criterion of ≥ 70% towards CR, that is, Proximal–Short (4/4 fish), Proximal–Long (3/4 fish), and Distal–Short (1/4 fish), reorientation within the non-visual geometric frame despite the visual 3D landmark emerged before learning.

The total choices (𝑓) made by fish towards the two diagonals (correct (CR), incorrect (NF)) were analyzed to evaluate the use of the non-visual geometric frame (rectangular transparent arena) alone, despite the visual 3D landmark (blue cylinder), by unsuccessful fish at reorienting. The three experimental conditions in which fish did not learn were collapsed; mean performance is reported. Results are shown in Figure 5b.

A repeated-measures ANOVA was performed by considering the total choices (𝑓) and including in the analysis eight fish (1 in PL, 3 in DS, 4 in DL). The ANOVA with sessions (all twenty-five training sessions) and diagonal (correct (CR), incorrect (NF)) as within-subject factors, and condition (PL, DS, DL) as between-subject factor, showed the following results: there was a significant effect of session (*F*_(24,120)_ = 2.67, *p* < 0.001, $\eta_{p}^{2}$ = 0.35) and of diagonal (*F*_(1,5)_ = 45.8, *p* = 0.001, $\eta_{p}^{2}$ = 0.9); there was no significant effect of session ∗ condition (*F*_(48,120)_ = 0.87, *p* = 0.71), diagonal ∗ condition (*F*_(2,5)_ = 0.54, p = 0.62), session ∗ diagonal (*F*_(24,120)_ = 0.63, *p* = 0.9), session ∗ diagonal ∗ condition (*F*_(48,120)_ = 0.86, *p* = 0.72), or condition (*F*_(2,5)_ = 3.76, *p* = 0.1).

In the experimental conditions where not all the fish met the learning criterion, that is, Proximal–Long (1/4 fish), Distal–Short (3/4 fish), and Distal–Long (4/4 fish), reorientation within the non-visual geometric frame despite the visual 3D landmark emerged during training.

### 3.5. Latency Times by Conditions (PS, PL, DS, DL) in the First and Last Training Sessions

The time spent (s) by fish within the experimental arena before going out (through a correct choice) was analyzed in the first and last training sessions to evaluate possible improvements in latency times due to proximity–length interactions. Results are shown in Figure 6.

A repeated-measures ANOVA was performed by considering the total amount of time (s) spent by fish within the experimental arena before going out. The ANOVA with session (the first and last training session) as the within-subject factor and condition (PS, PL, DS, DL) as the between-subject factor showed a significant effect of session (*F*_(1,12)_ = 109.41, *p* < 0.001, $\eta_{p}^{2}$ = 0.9), of session ∗ condition (*F*_(3,12)_ = 11.74, *p* < 0.001, = 0.75), and of condition (*F*_(3,12)_ = 11.07, *p* < 0.001, $\eta_{p}^{2}$ = 0.74).

Post hoc multiple comparisons with Bonferroni’s correction revealed significant differences between PS and DS (*p* = 0.007), PL and DS (*p* = 0.001), and PL and DL (*p* = 0.05), but not between PS and PL (*p* = 0.1), PS and DL (*p* = 0.37), and DS and DL (*p* = 0.32). As regards the interaction session ∗ condition, significant differences were found between the first PS session and the first DS session (*p* < 0.001); the first PL session and the first DS session (*p* < 0.001); the first PL session and the first DL session (*p* = 0.008); the first DS session and the last PS session (*p* < 0.001); the first DS session and the last PL session (*p* < 0.001); the first DS session and the last DS session (*p* < 0.001); the first DS session and the last DL session (*p* < 0.001); the first DL session and the last PS session (*p* < 0.001); the first DL session and the last PL session (*p* < 0.001); the first DL session and the last DS session (*p* < 0.001); the first DL session and the last DL session (*p* < 0.001). All the other interactions were not significant (*p* = 1).

In Distal–Short and Distal–Long conditions, fish decreased their time spent within the experimental arena before going out. However, Proximal–Short and Proximal–Long fish did not show such a tendency.

### 3.6. Motion Patterns by Collapsed Conditions (PS, PL, DS, DL): Strategy and Direction

The total choices (𝑓) made by fish towards the Correct were analyzed to evaluate if a consistent motion strategy (wall-following vs. center-to-corner) and motion direction (left vs. right) were used by fish once they had learned to reorient within the non-visual geometric frame (rectangular transparent arena) equipped with the visual 3D landmark (blue cylinder). Results are shown in Figure 7.

A repeated-measures ANOVA was performed by considering the total choices towards the Correct (C_W_ vs. C_C_) (𝑓) and including in the analysis all 16 fish (4 in PS, 4 in PL, 4 in DS, 4 in DL). The ANOVA with session (the sessions of learning and validation for successful fish and the last two training sessions, i.e., 24th and 25th, for unsuccessful fish) and strategy (wall-following, center-to-corner) as within-subject factors, and condition (PS, PL, DS, DL) as the between-subject factor, showed the following results: There was a significant effect of strategy (*F*_(1,12)_ = 4364.63, *p* < 0.001, $\eta_{p}^{2}$ = 1). There were no significant effect of session (*F*_(1,12)_ = 1.15, *p* = 0.3), session ∗ condition (*F*_(3,12)_ = 0.79, *p* = 0.53), strategy ∗ condition (*F*_(3,12)_ = 0.39, *p* = 0.76), session ∗ strategy (*F*_(1,12)_ = 2.01, *p* = 0.18), session ∗ strategy ∗ condition (*F*_(3,12)_ = 0.81, *p* = 0.51), or condition (*F*_(3,12)_ = 0.05, *p* = 0.98).

A repeated-measures ANOVA was performed by considering the total choices towards the Correct (C_WL_ vs. C_WR_) (𝑓) and including in the analysis all 16 fish (4 in PS, 4 in PL, 4 in DS, 4 in DL). The ANOVA with session (the sessions of learning and validation for successful fish and the last two sessions of training, i.e., 24th and 25th, for unsuccessful fish) and direction (Left, Right) as within-subject factors, and condition (PS, PL, DS, DL) as the between-subject factor, showed the following results: There was no significant effect of session (*F*_(1,12)_ = 1.64, *p* = 0.23), session ∗ condition (*F*_(3,12)_ = 0.79, *p* = 0.52), direction (*F*_(1,12)_ = 1.93, *p* = 0.19), direction ∗ condition (*F*_(3,12)_ = 1.15, *p* = 0.37), session ∗ direction (*F*_(1,12)_ = 0.3, *p* = 0.59), session ∗ direction ∗ condition (*F*_(3,12)_ = 0.28, *p* = 0.84), or condition (*F*_(3,12)_ = 0.16, *p* = 0.92).

Regardless of the experimental condition, in both the second last and the last training session, fish approached the Correct using a wall-following (perimetrical) motion strategy but without showing a preferential direction.

## 4. Discussion

This study explored whether zebrafish’s reorientation behavior could rely on two sources of spatial information, i.e., non-visual environmental geometry and a visual 3D landmark, which depend on two different sensory pathways, i.e., haptic/tactile-like stimulation and sight-dependent mechanisms. The global-shape information was provided by a rectangular transparent arena, and the conspicuous 3D landmark was provided by an outside blue cylinder. This landmark was located externally to the rectangular transparent surfaces, adjoining to the arena’s perimeter, and we varied its proximity to one target corner position (proximal or distal) and in relation to surface length (short or long side of the arena). Thus, four experimental conditions were used: in Proximal–Short (PS), the landmark was placed near the correct corner position (by now, “Correct”) on the short side of the arena; in Proximal–Long (PL), the landmark was placed near the Correct on the long side of the arena; in Distal–Short (DS), the landmark was placed far from the Correct on the short side of the arena; in Distal–Long (DL), the landmark was placed far from the Correct on the long side of the arena. Fish could distinguish the Correct from its geometric twin (“Rotational”) by conjoining the non-visual geometry of the arena (metric parameters: length, distance, corners; sense information: left, right) with the visual object breaking the spatial symmetry (e.g., a landmark on the short wall right for the Correct; no landmark on the short wall right for the Rotational) or through a cue-directed strategy.

We were interested in understanding whether zebrafish’s reorientation behavior could strictly depend on learning processes over time being finalized. In fact, a previous study by Lee and colleagues [32] showed that untrained zebrafish did not spontaneously reorient through a non-visual geometric frame equipped with a visual 3D landmark when their reorientation behavior was observed in a social-cued (working) memory task. By adapting the methods to our new purposes, a rewarded exit task [21,22,33,40,41,42,43,44,45,46] was carried out, with the aim of investigating possible effects of behavioral tasks, i.e., spontaneous choices—unrewarded; and training—rewarded, on zebrafish’s reorientation.

Results showed a range of difficulties in learning, which first depended on the proximity of the cylindrical landmark to the Correct. Half of the fish (8/16: 4/4 PS, ¾ PL, ¼ DS) learned to reorient with the transparent arena’s shape and the blue cylinder, and the other half (8/16: ¼ PL, ¾ DS, 4/4 DL) failed the task. However, in PS, PL, and DS conditions, fish chose the Correct corner position more than the incorrect ones, thereby using the cylindrical landmark when proximal and when distal but placed on the arena’s short side. With a chance level of 25%, even a performance around 55–60% is significant, but for our purposes, not enough to reveal proximity–length interactions, if any.

Since acquired reorientation behavior driven by non-visual geometry alone has been recently found in zebrafish (where fish confused the two symmetric corner positions diagonally placed: [22]), in the present study, the choices towards the two arena’s diagonals were also evaluated, regardless of learning. Results showed geometry use, i.e., more choices towards the geometric twin corners, by both successful and unsuccessful fish. It seems likely that the “coding” of shape parameters might have had primacy effects on landmark use, mostly when distal, anchoring reorientation behavior at the geometric level and preventing landmark use. Further studies should look into this issue, perhaps through a pre-training stage with geometry alone, followed by a subsequent training session with a landmark to better understand if overlapping codes affect reorientation behavior relying on spatial sources integration or occur at different time points during extensive learning. Additionally, this analysis would allow one to explore blocking and/or overshadowing effects [50,51,52,53].

Interesting differences were found between proximal and distal conditions, considering the mean number of learning trials and the latency times before going out from the arena. As such, PL fish needed a higher number of trials than PS fish, but both the groups were faster if compared with DS and DL fish, not showing a significant decrease in the time spent within the arena between the first and last training session. The absolute distance may explain this effect [54], along with interacting with the landmark distance. From the arena’s center, i.e., the starting position of the fish, to the landmark position, the absolute distance was minor when the visual cue was placed on the long side, and thus, in PL and DL conditions. Conversely, when placed on the short side, it was far from the center. However, the nearness of the blue cylinder seems to have been enough for learning in proximal conditions. It can be suggested that the recruitment of different processes (allo- and ego-centered) may affect long-term learning as regards the use of non-visual geometry and a landmark, at least in fish.

In this study, a geometric task based on rewarded training was performed, with the aim of highlighting possible differences compared to spontaneous choices. In contrast to Lee and colleagues [32], zebrafish learned to reorient within the non-visual geometric frame, i.e., the rectangular transparent arena, equipped with the visual 3D landmark, i.e., the blue cylinder, in both the proximal conditions (short/long). Trained zebrafish also overcame the attractiveness bias they spontaneously exhibit towards the closest position (i.e., the corner near the Correct) as a result of repeated experience. Previous evidence about the impact of behavioral methods on reorientation has suggested that training protocols aid fish in better representing spatial relationships within geometric layouts [22,40], and in efficiently using conspicuous and local landmarks to join target locations [46]. In appetitive contexts where motivational states acquire an increment over time, food and social incentives may trigger cognitive adaptations to face high-level demands in zebrafish, which are sensitive to these kinds of rewards [29,55,56].

As regards the use of exploratory motion patterns, in terms of strategy (wall-following vs. center-to-corner) and direction (left vs. right), results showed a non-lateralized preference for moving close to the arena’s perimeter, irrespective of the experimental condition. These findings add theoretical complexity to previous evidence (insects: [10,11]; chickens: [12,13,14]; simulated robots: [15,16,17,18,57]) but do not disclose the nature of the processing beneath. As such, it is unclear whether disoriented navigators build representations of environmental geometry and landmarks through view-matching processes (global: [10]; local: [14]), general associative rules [15], motor action sequences [17,18,57], modular codes [9], or haptic/tactile-like mechanisms [19,20,21,22]. Currently, the preponderance of evidence from spatial reorientation research converges on what animals can do but not on how. We suggest that, in general, a kind of multisensory “code” may govern reorientation behavior, where species-specificities determine what specific-domain processes are mostly involved. In fish species, both sight and haptic/tactile-like modalities may work in sync and become even more precise over consistent experience driven by motivational states. Moreover, our findings may support the view of an “amodal core-system of geometry” [58,59], where basic intuitions blossom out irrespective of visual stimulation.

## 5. Conclusions

Spatial reorientation driven by non-visual and visual stimulation was found in zebrafish, where proximity–length interactions led to an increasing difficulty in solving the spatial symmetry problem (i.e., distinguishing correct from rotational corner positions). However, geometry use of transparent surfaces clearly emerged irrespective of learning in all the experimental conditions, strongly supporting geometry use even under environmental visual deprivation. Since wall-following exploration was found in any case, whatever the experimental handling, it could be assumed that such a strategy did not determine learning; however, it could have enhanced global-shape representation (e.g., move about the periphery to determine a reference frame from which to build new spatial relationships).

Further studies should critically address the impact of multisensory exploratory routines on reorientation behavior and could involve brain-centered analyses focused on biochemical and neurobiological processes. As such, the use of zebrafish as a model would provide the multidomain background needed to pursue the goal.

## Figures and Tables

**Figure 1 animals-13-00440-f001:**
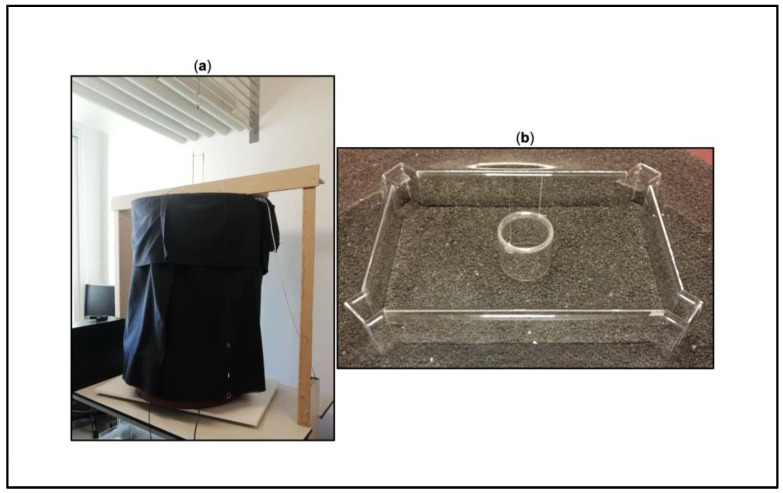
Experimental apparatus. (**a**) Setup from the outside. (**b**) Rectangular transparent arena with upright corridors at the corners. The transparent cylinder hosting the experimental fish at the beginning of each training trial is also shown (credit by Sara Boffelli).

**Figure 2 animals-13-00440-f002:**
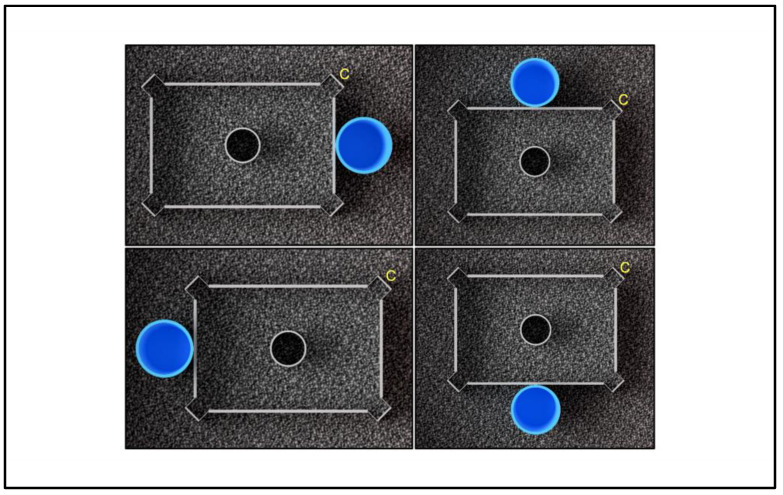
Experimental conditions. Proximal–Short (PS), top left; Proximal–Long (PL), top right; Distal–Short (DS), bottom left; Distal–Long (DL), bottom right (3D virtual graphic reconstruction by Sara Boffelli).

**Figure 3 animals-13-00440-f003:**
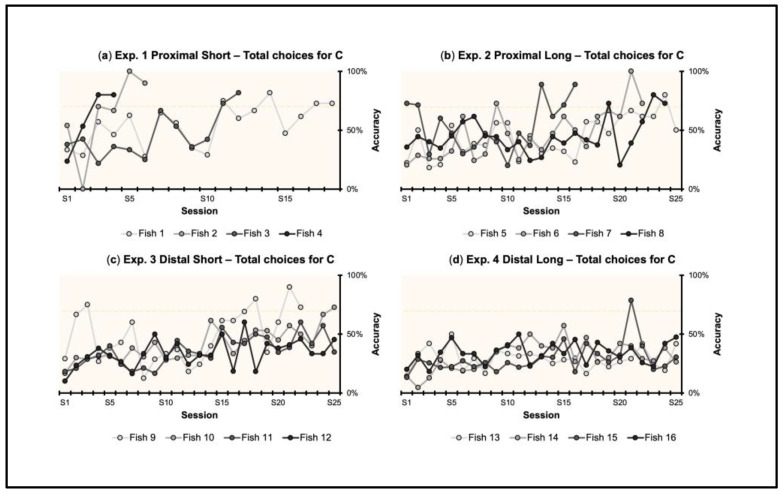
Individual learning curves in the four experimental conditions (PS, PL, DS, DL): total choices. In PS (**a**), 4/4 fish met the 70% accuracy criterion. In PL (**b**), 3/4 fish met the criterion. In DS (**c**), 1/4 fish met the criterion. In DL (**d**), 0/4 met the criterion. The dotted line indicates the 70% threshold.

**Figure 4 animals-13-00440-f004:**
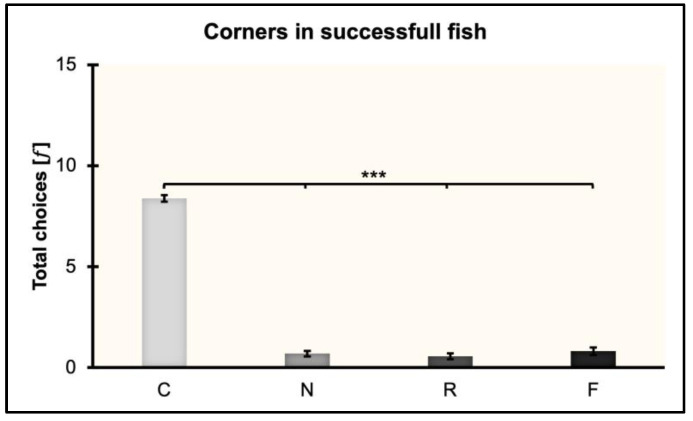
Total choices for the corners (geometry and landmark) in PS, PL, and DS collapsed. Mean ± SEM are shown. Significant *p* values are indicated with asterisks (*** *p* < 0.001).

**Figure 5 animals-13-00440-f005:**
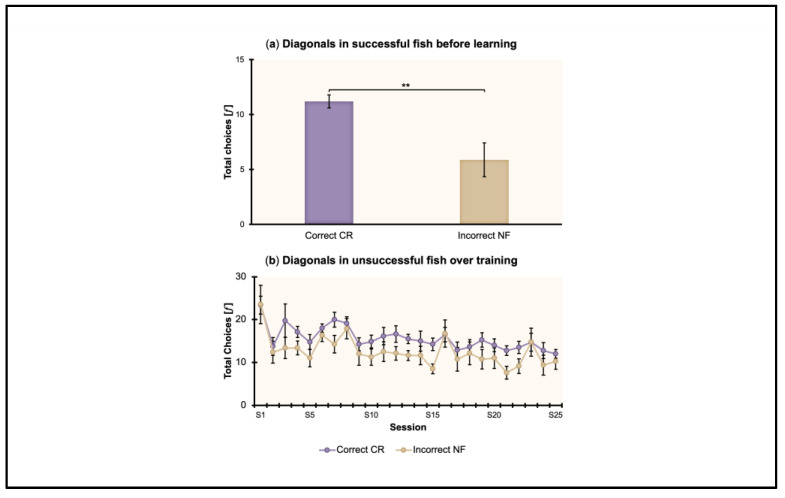
Total choices for the diagonals (geometry alone) in the 8 fish achieving the learning criterion ≥ 70% for C (**a**) and in the 8 fish not achieving it (**b**). Mean ± SEM are shown. Significant *p* values are indicated with asterisks (** *p* = 0.001).

**Figure 6 animals-13-00440-f006:**
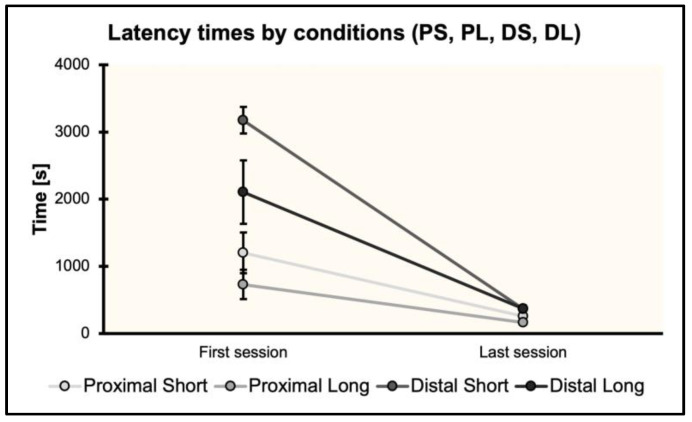
Latency times before leaving the arena through a correct choice, in the four experimental conditions (PS, PL, DS, DL) separately.

**Figure 7 animals-13-00440-f007:**
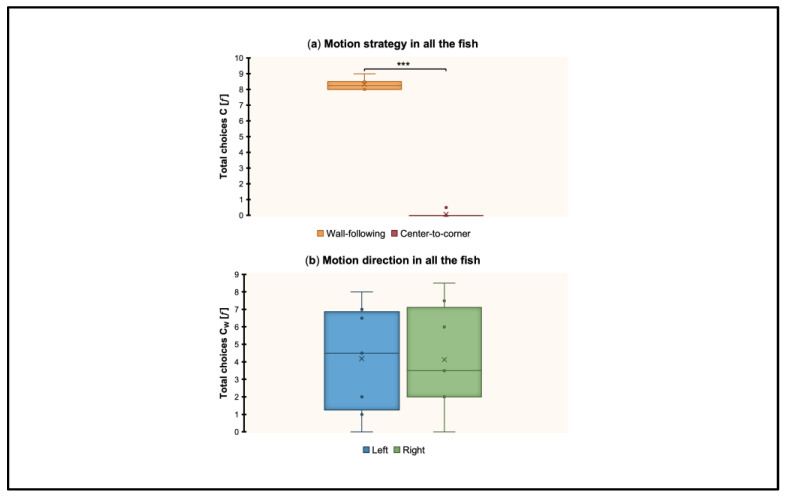
Motion patterns by collapsed conditions (PS, PL, DS, DL). (**a**) Motion strategy: wall-following vs. center-to-corner. (**b**) Motion direction: left vs. right. Mean ± SEM are shown. Significant *p* values are indicated with asterisks (*** *p* < 0.001).

**Table 1 animals-13-00440-t001:** First choices (%) in both the learning and validation sessions.

Fish	Condition	% Correct Learning	% Correct Validation
1	PS	75	75
2	PS	100	87.5
3	PS	75	75
4	PS	75	75
6	PL	100	75
7	PL	66.67 ^1^	87.5
8	PL	87.5	75
9	DS	87.5	87.5

^1^ Fish #7 (PL) achieved 66.67% in the learning session but highly improved its accuracy in the subsequent validation session.

## Data Availability

Not applicable.

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
