# Peer review of "Spatial Learning by Using Non-Visual Geometry and a Visual 3D Landmark in Zebrafish (*Danio rerio*)"

_animals, 2023, doi:10.3390/ani13030440_

Round 1
Reviewer 1 Report
Review Report
The present study analyzes for the first time the ability of zebrafish of conjoin non-visual, haptic/tactile-like stimulation and sight-dependent information when trained to locate a reinforced goal in a spatial reorientation task. The study is coherently framed in a consistent and sustained line of research and constitutes a forward step in the analysis of the multisystemic integration mechanisms governing reorientation behavior. Considerations of authors regarding the nature of the reorientation tasks (mainly, spontaneous vs rewarded) are of critical importance to understand how spatial behavior may be affected, or not, by an integrative multisensory experience.
General concept comments
However, a major concern arises after a detailed reading of the paper regarding a critical methodological aspect. Specifically, authors claim that zebrafish (at least, those in the PS, Pl and DS conditions) learn to conjoin tactile-like information (provided by a transparent rectangular arena) and visual information (provided by a blue cylinder) to chose the “Correct” exit, but this statement is no supported by the results. If an alternative hypothesis were have been taken (for instance, fish learn to locate the Correct based on a cued-directed strategy, that is, using the blue cylinder), the results from the current data analysis will continue supporting the new hypothesis. So, ¿how and why have this possibility been dismissed? ¿On what precise data analyses are based the hypothesis of multisensory integration proposed by the authors? In fact, results in the epigraph 3.5., authors claim that “In the experimental conditions where almost all the fish met the learning criterion ≥ 70% towards the Correct, that is, Proximal Short (4/4 fish), Proximal Long (3/4 fish), and Distal Short (1/4 fish), the use of non-visual geometry alone did not emerge”, suggesting that if they were successful fish and, therefore, they learned to locate the Correct, they did it by using the blue cylinder. In other words, this study lacks of the appropriate experimental design as to state what is stated and to reject an alternative hypothesis. Definitively, control groups or specific probe trials for the fish in the present study are highly required.
Minor concerns also arise while reading the manuscript. Regarding the use of statistical tools. Repeated measures ANOVAs depending on intra- and extra- factors are extensively used in the present study to compare values of the different variables. Not always ANOVAs are well driven. For instance, there are two cases in this study in which when an interaction between factors results statistically significant, the post-hoc multiple comparisons are not analyzed (Corner x Condition interaction in which should to be the epigraph 3.2., and Sessions x Condition interaction in epigraph 3.6). Why not? Once an ANOVA use is decided to be done, this is the indicated way to study the interdependency between factors. Even more, authors use to apply another statistical tool (Student’s t-Test) for this purpose. Authors should justify the rejection of the multiple comparisons-based analyses and its substitution by the Student’s t-Tests. In this line, another abnormal use of ANOVA rises in epigraph 3.4. (Diagonals by conditions (PL, DS, DL) collapsed, over 25 training sessions). No Sessions x Diagonal interaction was obtained from the ANOVA, which automatically precludes the analysis of pot-hoc multiple comparisons. However, authors performed a paired samples t-Test to look for significant differences between CR and NF in two sessions in a row throughout the 25 sessions. Exactly the same occurs in epigraph 3.5. (Diagonals by conditions (PS, PL, Ds) collapsed), where Sessions x Diagonal interaction is not significant and “a paired samples t-test was performed to 360 analyze if the difference between CR and NF was consistent in both the sessions”. Again, in these two cases, the authors should justify the use of the Student’s t-Tests.
A delicate issue is that regarding the number of experimental animals included in each one of the statistical analyses. In no case is the number of fishes explicitly identified. By defect, one can assume a number of 4 for each condition considered, but it is just an assumption. This issue is important for all the analyses carried out in this study, but is particularly severe in those of them in which successful and unsuccessful fish were studied (analyses in the epigraphs 3.3., 3.4., and 3.5.). In these cases, what fishes are supposed to were included in each condition, all of them or only those who reached the learning criterium? For example, in analysis 3.5., conditions PS, PL, and DS are included. Specifically for DS, how many animals were included for the analysis, all of them or just that one who learned the task? This point is extremely important both methodologically and conceptually considered, but no word in this regard is explicitly said. Authors should appropriately and punctually inform about this point in the manuscript.
Other minor consideration arises regarding the criterium to assign values to the “Sessions” factor in the ANOVA, particularly from analysis 3.2. to 3.5. The last 4 sessions in 3.2., then 2 sessions (learning and validation) in 3.3., the whole 25 sessions in 3.4., and finally and surprisingly, 2 sessions (the last 2 before learning) in 3.5. All these changes in the adopted criterium look like arbitrary and interested or, in any case, are not well justified. Authors should better justify this situation.
Another consideration regarding the number of sessions. In the analysis of the epigraph 3.4., a repeated measures ANOVA with Sessions (all the twenty-five training sessions) and Diagonal (CR, NF) as within-subject factors, and Condition (PL, DS, DL) as between-subject factor was performed. Four of the twelve “unsuccessful” animals (3 in PL and 1 in DS) reached the learning criterium before session 25. That tis, not all the animals had 25 sessions. Briefly, how were the gaps (lost data) for these four animals filled when performing the repeated measures ANOVA?
Taken together, all these methodological issues (the weakness of the experimental design, the careless use of the statistical tools, and the apparent arbitrariness of some criteria in the analysis), leads to the consideration that conclusions in this study are not supported by data and therefore, that we are still not able to affirm that fish may combine multisensory information to reorientate in the space.
Apart from methodological considerations, it has to be noted that mostly the cited references are not particularly recent and many of them are auto-citations. To updated references should be recommendable.
Specific comments
Introduction.
Line 90: Please, a brief explanation of "passive disorientation" is recommended.
Materials and Methods.
Line 113: Reference must be a number in square brackets.
Line 117: For a non-familiarized person, the term “corridor” leads to confusion since a corridor use to stay horizontal and not vertical, as is the case in this procedure. Please, be more accurate in the description of corridors and its attachment with the central arena.
Fig.1: Representation of corridors in (c) looks too decontextualized from the central arena. In fact, for a non-familiarized person it is hard to integrate with the main arena.
Results.
Line 211: What does "above-chance performance" mean, above 25% or 70%? Please, do make clearer this point.
Fig. 3: It would be appreciated if the learning criterium (≥ 70%) threshold were showed.
Line 218: Epigraph “3.2.” instead of “3.1.”.
Fig. 4: Not absolutely certain on that but, it is needed to write all the significant values in the footnote of the figure, even in the case that some specific significant values are not represented on it? Also applicable to Fig. 6 and Fig.8.
Line 291: The result depicted in this little paragraph do not relate with the concept expressed in the epigraph 3.3. (Corners by conditions (PS, PL, DS) collapsed). Please, think about a better relocation for it or create another specific epigraph for it.
Table 1: There must be a mistake in the annotation at the foot of the table. It more likely to be Fish #7 instead of Fish #3.
Line 369: ¿"... a correct choice" (CR) or "... the correct choice" (C)? Please, make it clear.
Line 401: Please make clear if “Second last session” and “Last session” are respectively the “Learning” and “Validation” sessions.
Reviewer 2 Report
This paper investigates reorientation behaviour of zebrafish in a transparent arena with a 3D landmark. Conjoining non-visual geometry and visual landmark for rewarded reorientation is available and can be affected by proximity-length interactions. The manuscript is well written and the analysis and conclusion are well presented. The objectives and contents are clear and the paper is innovative. Thus, I would like to see the paper published in this journal. The author may want to address the following comments for potential improvement.
In each training session, are all four fish put together or individually in the arena? Is it possible to test 16 fish in each session? 4 might be a small number regarding the individual difference between zebrafish.
There are 25 training sessions but the authors also mentioned “two sessions in a row (learning and validation)”. Are learning and validation a single training session? Or there are 13 learning sessions and 12 validation sessions?
Less than 7 sessions were tested/illustrated for Fish 2 and 4 in Fig. 3(a). And the Y-axis of Fig. 3(a) is S20 rather than S25. Is the test terminated when met the 70% accuracy criterion? If so, the accuracy of Fish 9 in distal short (Fig. 3c) is more than 70% in S3 and drops below 50% in S4 – S14.
Round 2
Reviewer 1 Report
After reading the second version of the manuscript I consider that included changes improved the original one. Therefore, I accept the manuscript in its present form.
In any case, I make a final and critical consideration regarding the experimental control mechanisms. Unfortunately, the work still continues lacking of enough strength as to establish robust causal relations among the intervenient variables despite the authors’ endeavored to reduce the forcefulness of their original conclusions about “conjoining” in this second version. I particularly appreciate it, but for future studies, I highly recommend the use of stronger experimental designs including control groups and probe trials. Please, consider this respect seriously.
Thanks to the authors for kindly considering my comments and suggestions and for including all the necessary changes proposed.
Finally, let me point out some final considerations:
Consider an explicit allusion to Bonferroni's Test when referring to the ANOVAs in the last paragraph of Methods.
There is a mismatch mistake in Fig.5. Please, commute the positions of charts for successful and unsuccessful animals in order to be congruent with the information for these groups in both the text and the figure caption.
Please, consider to enlarge some figures, particularly figures 5 and 7.
